# Machine Learning Applied to LoRaWAN Network for Improving Fingerprint Localization Accuracy in Dense Urban Areas

Andrea Piroddi [1,*,†] and Maurizio Torregiani [2,†]

1    Department of Computer Science and Engineering, Università di Bologna, 40126 Bologna, Italy
2    Telebit S.p.A., 31030 Casier, Italy
*    Correspondence: andrea.piroddi@unibo.it
†    These authors contributed equally to this work.

**Abstract:** In the area of low-power wireless networks, one technology that many researchers are focusing on relates to positioning methods such as fingerprinting in densely populated urban areas. This work presents an experimental study aimed at quantifying mean location estimation error in populated areas. Using a dataset provided by the University of Antwerp, a neural network was implemented with the aim of providing end-device location. In this way, we were able to measure the mean localization error in areas of high urban density. The results obtained show a deviation of less than 150 m in locating the end device. This offset can be decreased up to a few meters, provided that there is a greater density of nodes per square meter. This result could enable Internet of Things (IoT) applications to use fingerprinting in place of energy-consuming alternatives.

**Keywords:** IoT; localization; LoRaWAN; deep learning





## 1. Introduction

The growing interest of the telecommunications market for IoT technologies is driving research to develop different Low-Power Wide-Area Network (LPWAN) standards. Simply put, LPWAN should be to IoT what WiFi is to consumer networks. Base stations provide (very) wide radio coverage and adjust transmit power, transmit rate, modulation, duty cycle, etc., so that end devices experience little energy consumption from the connection [1]. Ultralow power consumption, as well as ubiquitous outdoor and indoor connectivity, are fundamental aspects to ensure that the network of IoT devices is reliable over the years. To ensure smooth operations on IoT networks, it is necessary to take into account various elements such as network topology, modulation techniques, complexity of the hardware, the use of the radio spectrum and regulations [2]. From this point of view, it follows that context awareness is a key element in IoT applications [3]. In order to set up context awareness, device location should be determined with minimal location error. The simplest way to achieve this would be to use the GPS tracker. Unfortunately, however, the GPS receiver can consume up to 50 mA when detecting the position [4]. Added to this is the fact that, once the position has been obtained, it is necessary to transmit it to the gateway, and this further step produces an additional energy consumption. A further element to consider is the high accuracy of a GPS measurement, an accuracy that is often not required in an IoT sensor network. So, in the face of a high energy consumption, we would have an excessively detailed measure in the context under analysis. A particularly interesting technique has been developed in [5], in which a simplified implementation of interferometry is presented, obtaining high accuracies. This technique does not require additional hardware, but it cannot be implemented with all communication devices, as it strongly depends on the modulation mechanisms used and the freedoms left to users. Wireless localization based on LPWAN communication is therefore an interesting alternative for localization in low-power

networks [6]. These techniques estimate transceiver location by analyzing the physical characteristics of the transmission link, such as received signal strength (RSS) values, packet arrival time information and so on [7]. This paper aims to verify if, by applying deep learning methodologies to fingerprint techniques, it is possible to obtain interesting results in terms of minimizing the localization error. The performance of fingerprint-based methods depends on the number of reference points (RP) per room unit. However, since RSS measurements are onerous and time-consuming, increasing the number of RPs increases the positioning cost [8]. The remainder of the paper is structured as follows. Section 2 describes the LoRaWan standard used to collect the dataset. Section 3 shows how the dataset has been built. Section 4 illustrates the machine learning approach that we used to estimate the location of end devices. Section 5 shows the results of our technique. In Section 6, we discuss these results. Finally, Section 7 shows the conclusions and the intended future work.

## 2. LoRaWAN Standard

LoRaWAN technology provides two-way communication, but the transmission from node (also known as end devices) to gateway (also known as concentrator or base station) or uplink message is the most frequent one, compared with that from gateway to node or downlink. This is due to the purpose of the nodes, which is to collect data and then send them to the Network Server. Lastly, data will be sent to the Application Server.

Nodes send radio frequency uplink messages to the gateway using LoRa modulation. The gateway forwards the message to a network server over an IP connection routed via Ethernet, Wi-Fi, or 3/4/5G, adding information about the quality of the communication.

The nodes send messages in uplink to all gateways in their transmission range in broadcast mode. The Network Server takes care of the management of duplicate uplink messages and the selection of the best gateway to use if a downlink message is to be sent to the node.

The Network Server also manages the bit rate of the nodes through the ADR (Adaptive Data Rate) mechanism to maximize the network capacity and extend the battery life of the node. For example, the TTN Network Server uses the 20 most recent uplink messages, starting from the moment the ADR bit is set, to determine the optimal data rate [9]. These measurements include the number of frames, signal-to-noise ratio (SNR) and number of gateways that collected each uplink message.

The Application Server instead takes care of receiving and analyzing the data sent by the nodes and determining the actions that must be performed by the nodes.

LoRaWAN is based on CSS (Chirp Spread Spectrum) technology [10]. Chirps (also known as symbols) are the data carrier. The Spread Factor (SF), i.e., the number of bits in a chirp, allows to control the length of the chirp and thus to control the data transmission rate [6]. The lower the spreading factor, the faster the chirp and the higher the data rate. Every time the spreading factor is doubled, the chirp sweep rate and data transmission rate are halved [10]. In this paper, we do not address the trade-off between the SF, bit rate, network coverage and energy consumption. LoRaWAN uses the ISM (Industrial, Scientific and Medical) frequency bands reserved for noncommercial radio communication applications, such as industrial, scientific and medical use. In particular, depending on the geographic area and related regulations, the two most common frequencies are 868 MHz in Europe and 915 MHz in North America. Figure 1 shows a picture of the LoRaWAN network architecture.

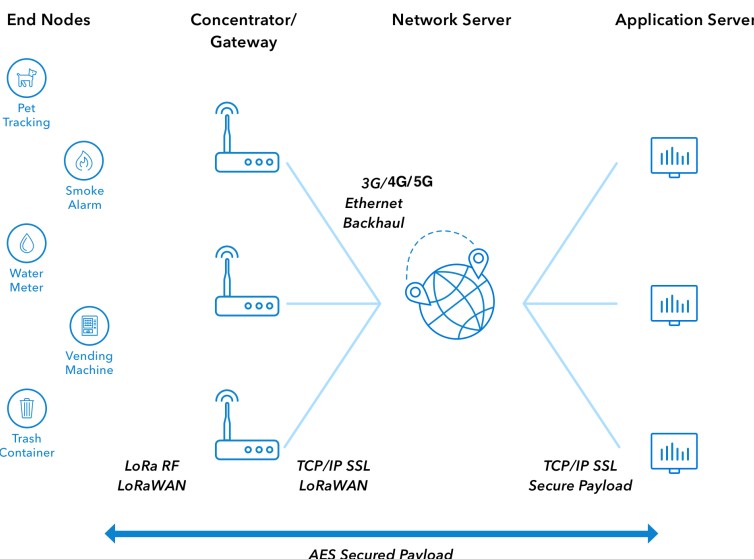

**Figure 1.** LoRaWAN network architecture.

## 3. Dataset Analysis

In the period between November 2017 and February 2018, the dataset on which our study is based was collected by the Faculty of Applied Engineering of the University of Antwerp [11]. Hardware consisting of a GPS receiver and LoRaWAN end device was installed on about twenty cars from the Antwerp postal service. While the 20 cars drove around in the city center, the location information was sent in a LoRaWAN message. A callback function was configured on the LoRaWAN backend server to forward the payload of each message along with additional network information to the local data server. In the dense urban area explored, 72 LoRaWAN gateways were detected. Figure 2 shows a picture of the Antwerp Urban Area and message locations.

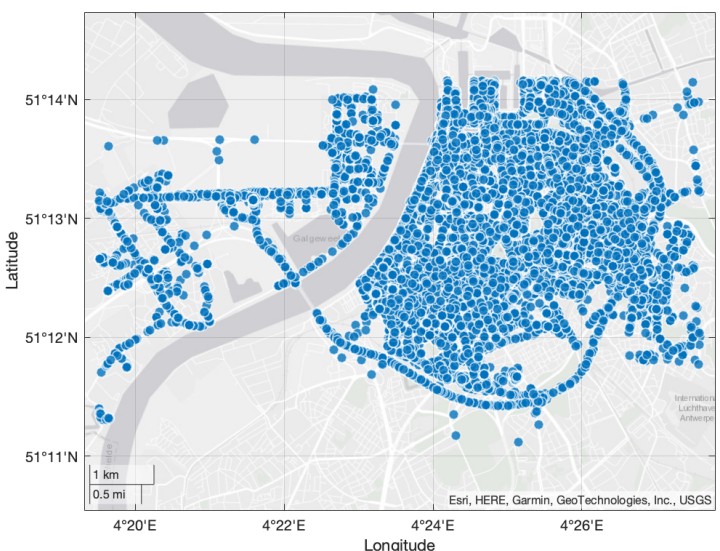

**Figure 2.** LoRaWAN dataset collected in Antwerp city center.

The urban dataset contains 130,429 messages and is available at [12]. Every record represents one of the 130,429 messages collected in the urban dataset, see Table 1. In the last four columns appear the receiving time, the spreading factor, latitude and longitude of a message. The previous column shows which of the 72 base stations in the metropolitan area received the message. An RSSI of −200 dB is reported in the cell if the base station does not receive the message. Received Signal Strength Indicator (RSSI) is a measure of the

power present in the received radio signal. This requires no additional power, hardware, or bandwidth. These properties of RSS measurements make them relatively cheap and easy to implement, making this technique attractive [13]. In previous works, a basic kNN fingerprinting localization technique was used. A parameter sweep was analyzed, varying *k* from 1 to 15. Considering that the optimal value of *k* was the one which produces the lowest mean location error, it emerged that the optimal value for *k* was 11 nearest neighbors. Applying this value of *k*, the LoRaWAN dataset returned a median error of 273.03 m and a mean error of 398.4 m.

**Table 1.** Structure of the urban LoRaWAN dataset. Each row is a LoRaWAN message showing the receiving base station (BS) with RSSI value, message reception time (RX time), LoRa spreading factor, latitude and longitude at time of transmission.

| BS 1 | BS 2 | ... | BS 72 | RX Time | SF | Latitude | Longitude |
|------|------|-----|-------|---------|-----|----------|-----------|
| −200 | −200 | −200 | −200 | "2019-01" | 8 | 51.23399... | 4.42610... |
| −200 | −118 | −200 | −97 | "2019-01" | 7 | 51.20718... | 4.40368... |
| ... | ... | ... | ... | "..." | ... | ... | ... |

By reproducing the same approach, it is possible to find the matrix of the centroid locations. Plotting the centroids on a map, as in Figure 3, it emerges that the resolution achievable with this technique is exactly the one indicated above. Calculating the distance of the centroids in the most densely urbanized area, it can be seen, in fact, that the values are around 796 m. This result confirms what we have seen so far, namely that in this densely urban context, using the kNN resolution technique, resolution does not drop below 398 m.

Is it possible to obtain a mean location error lower than that obtained with the kNN technique using the dataset as input in a neural network designed for deep learning attribute data classification? With the aim of answering this question, we started with a detailed analysis of the densely urbanized area. This area, in the specific case of Antwerp, is contained in a rectangle whose vertices have coordinates between [4°20′ East, 4°27′ East] and [51°11′ North, 51°15′ North]. We therefore divided the area subtended by the rectangle into subareas. Within each of these subareas, we can place a subset of the original dataset.

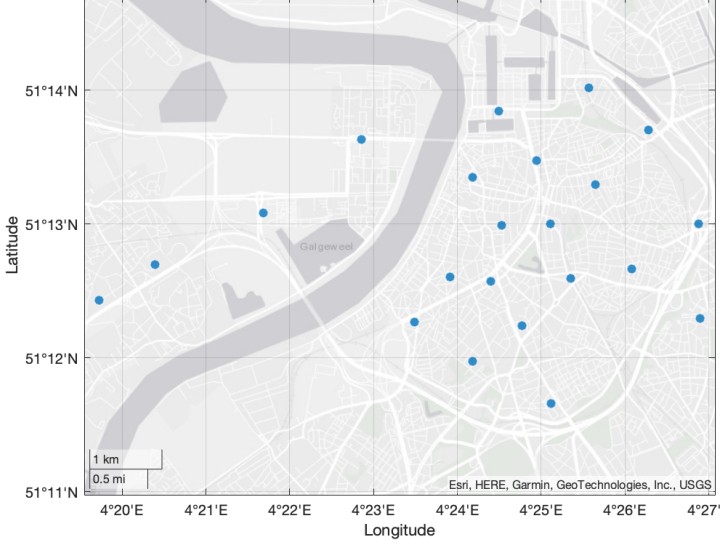

**Figure 3.** Geometrical Medians of Clusters obtained with kNN Technique.

This allows us to reconstruct the dataset with a further column whose information is related to the subarea which our message belongs to. For each of these subregions, we used a distance-based clustering function for a set of (*x*, *y*) coordinates [14]. The function returns clusters in a set of spatial points represented by (*x*, *y*) coordinates. The grouping operation

is based on the distance between points and does not require knowing the number of groupings in advance. Then, if the distance between two points is less than a custom threshold, for example 150 m, each point is clustered with its nearest neighbors.

The function returns basic summary statistics, such as the number of clusters, maximum, minimum and the average cluster sizes, as well as numbers showing the clusters, see Figure 4, the centroid points, see Figure 5, and the geometric median points of each groupings. It also yields two text files that contain the coordinates of all centroid points and geometric median points. For every cluster, the output variables return the $(x, y)$ coordinates of the geometric median point and of the centroid, as well as the $(x, y)$ coordinates of each point that composes the cluster. See an example in Table 2.

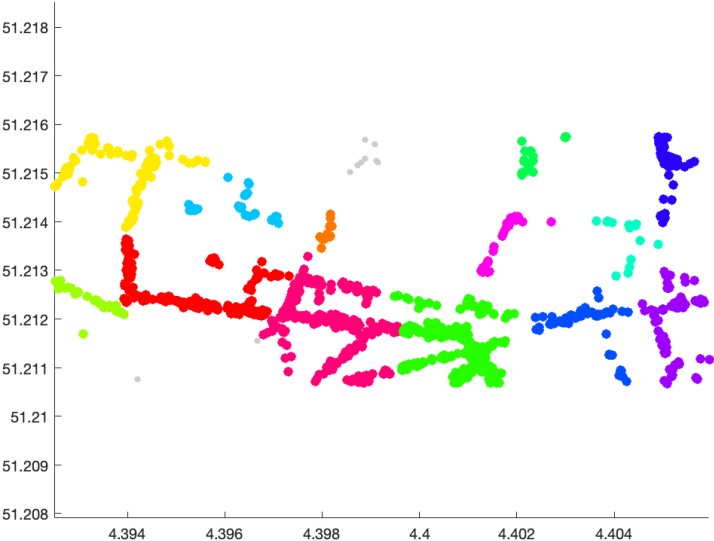

**Figure 4.** This figure shows the clusters obtained with the function for distance-based clustering of a set of XY coordinates in the coordinate range $Lat \in [4°23'00'', 4°24'00'']$–$Long \in [51°12'00'', 51°13'00'']$, which represents the subarea corresponding to the central area of the city of Antwerp.

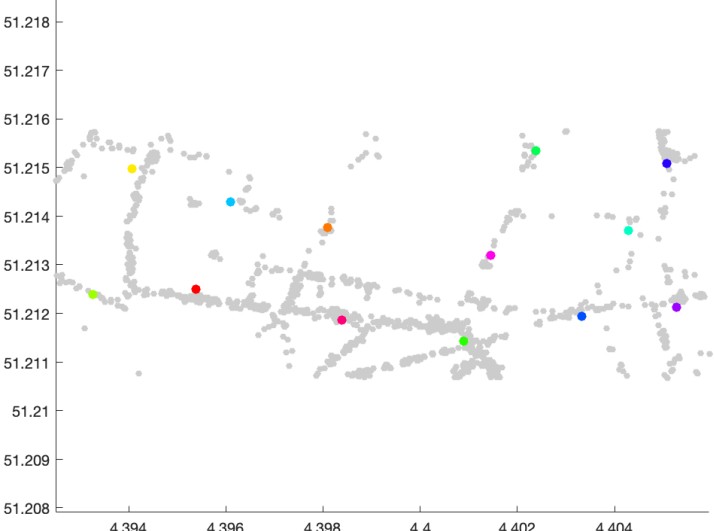

**Figure 5.** This figure shows the centroids of clusters obtained in the central Area of Antwerp: (see Figure 4) $Lat \in [4°23'00'', 4°24'00''] - Long \in [51°12'00'', 51°13'00'']$.

**Table 2.** Output of the distance-based clustering function in the central area of Antwerp: *Lat* ∈ [4°23′00″, 4°24′00″] − *Long* ∈ [51°12′00″, 51°13′00″].

| Number of Clusters | Size of Smallest Cluster | Size of Largest Cluster | Mean Cluster Size | Median Cluster Size | Number of Points Not Part of Any Cluster |
|---|---|---|---|---|---|
| 13 | 14 | 880 | 212 | 78 | 10 |

In [15], location experiments were performed in outdoor locations near a building. The Path Loss Factor (PLF) results show that it is possible to achieve good performance even in the presence of errors in the intersensory measurements.

## 4. Machine Learning Approach

Deep neural networks rely on large amounts of high-quality data to achieve satisfactory performance. Data quantity and quality are very important when training large and complex architectures. This is because deeper models typically have huge parameter sets that need to be trained and configured. This issue still applies to mobile network applications [16]. Using the MATLAB suite, we then configured a neural network that would take as input the RSSI data received from each of the 72 base stations, the values of the coordinates (latitude and longitude) of the message and the categorical value of the assigned subarea. This operation was recursively applied to all the subareas. To train a network using categorical features, we had to first transpose the categorical attributes to numeric using the ***convertvars*** function by defining a string array which contains the names of all the categorical input variables. The dataset was split into training, validation and test datasets, with 15% of the data used for validation and 15% for testing. The neural network was defined with a feature input layer (BS1, BS2, ..., BS72, lat, long and subarea), normalizing the data using the Z-score method. Then, a fully connected layer with an output size of 70 was added, followed by a batch normalization layer and a ReLU layer. Another fully connected layer with an output size equal to the number of classes was specified for classification, followed by a softmax layer and a classification layer, see Algorithm 1.

In Figure 6, the architecture of the neural network used in this work is shown.

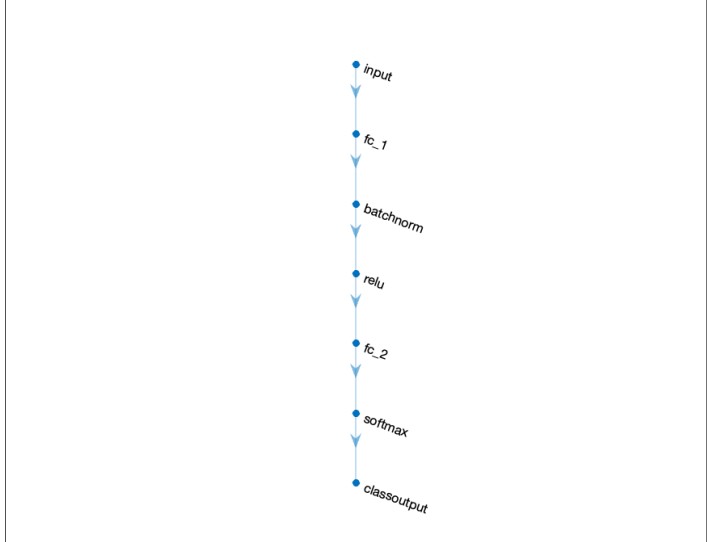

**Figure 6.** Architecture of the Neural Network.

---

**Algorithm 1** Neural Network Architecture and Training Options.

---

```
layers = [
    featureInputLayer(numFeatures,'Normalization', 'zscore')
    fullyConnectedLayer(70)
    batchNormalizationLayer
    reluLayer
    fullyConnectedLayer(numClasses)
    softmaxLayer
    classificationLayer];

miniBatchSize = 20;
options = trainingOptions('adam', ...
    'InitialLearnRate',0.0007,...
    'MiniBatchSize',miniBatchSize, ...
    'Shuffle','every-epoch', ...
    'ValidationData',tblValidation, ...
    'Plots','training-progress', ...
    'Verbose',false);
```

---

In Table 3, a comparison of localization precision that can be achieved with various available techniques is shown.

**Table 3.** Comparison of different techniques for localization.

| Technique | Range (m) | Power | Precision (m) | Source |
|-----------|-----------|-------|---------------|--------|
| OwLL | 500 | Low | ≈9 | [17] |
| Prior LP-WAN | >500 | Low | >100 | [18,19] |
| Sensor-based | ≈50 | Low | ≈5 | [20,21] |
| Cellular | ≈50 | High | 0.085 | [22,23] |
| Wi-Fi | ≈15 | Medium | <0.05 | [24,25] |

Using Table 3, we can see that the closest solution to the desired accuracy goal (with the same power) is the sensor-based approach. Therefore, the next step is to compare the neural network approach with the sensor-based one regarding the aspects of power consumption and computational complexity.

*4.1. Comparative Analysis between Computational Complexity of Neural Network and Sensor-Based Approach*

Ref. [26] shows that the RTT delay for messages sent from the end device to the controller and for responses returned from the controller to the end device over LoRaWAN ranges from 1.12 s to 6.59 s. Considering that, once the neural network has been trained, the response time is of the order of a few milliseconds [27]; therefore, the overall delay depends on the propagation (RTT) and not on the neural network architecture.

In [28], it is shown how the computational complexity of a neural network can be calculated. Looking at the inference part of a feedforward neural network, we have forward propagation.

Finding the asymptotic complexity of forward propagation can be performed similarly to how we found the runtime complexity of matrix multiplication.

Assume the input vector can be written as $x \in \mathbb{R}^n$.

The input is treated like any other activation matrix and has the following indices: $x = w^{(0)}$, the zeroth element, and $w_0^{(0)}$ is a bias unit with a value of 1, as usual. Forward propagation can be written as (see Figure 7).

$$z^{(k)} = \theta^{(k)} w^{(k-1)} \tag{1}$$

$$z^{(k)} \in \mathbb{R}^{1 \times (m|\theta^{(k)} \in \mathbb{R}^{m \times n})} \tag{2}$$

$$w^{(k)} = g(z^{(k)}) \tag{3}$$

From where $g(x)$ is the activation function, which is evaluated element-wise. So, we know $w^{(k)}$ are the same dimensions as $z^{(k)}$. We can see that the matrix multiplication and activation functions for each layer are computed. From [28], the asymptotic execution time for simple matrix multiplication is $O(n^3)$; since $g(x)$ is an element-wise function, the execution time is $O(n)$. Analyzing the dimensionality of the feedforward neural network reveals the following:

$$\theta^{(0)} \in \mathbb{R}^{n^{(0)} \times 1} \tag{4}$$

$$\theta^{(1)} \in \mathbb{R}^{n^{(1)} \times n^{(0)}} \tag{5}$$

$$\theta^{(2)} \in \mathbb{R}^{n^{(2)} \times n^{(1)}} \tag{6}$$

More generally:

$$\theta^{(k)} = \begin{cases} \mathbb{R}^{n^{(k)} \times 1}, & \text{if } k = 0 \\ \mathbb{R}^{n^{(k)} \times n^{(k-1)}}, & \text{if } k > 0 \end{cases} \tag{7}$$

where $n^{(k)}$ is the number of neurons containing bias units in the layer $k$.

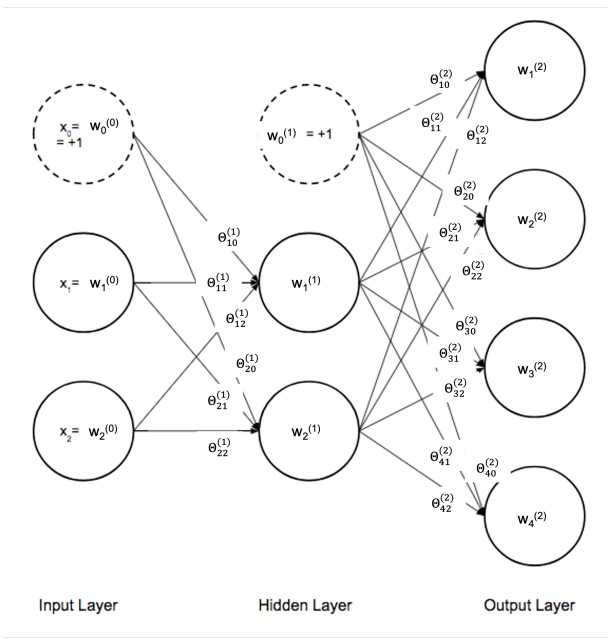

**Figure 7.** Neural Network Layers.

From [28], we know that

$$w = n^{(k)} \tag{8}$$

$$z = n^{(k-1)} \tag{9}$$

$$d = n^{(k-2)} \tag{10}$$

From this, we can see that:

$$n_{mul} = \sum_{k=2}^{n_{layers}} (n^{(k)} n^{(k-1)} n^{(k-2)}) + (n^{(0)} n^{(1)} 1) \tag{11}$$

$$n_g = \sum_{k=1}^{n_{layers}} (n^{(k)}) \tag{12}$$

where $n_{mul}$ is the number of multiplications performed and $n_g$ is the number of times the activation function $g$ is applied.

Note, however, that we assumed the following approximation, that is, $w^{(k)}$ has the same dimensions as $\theta^{(k)}$, but clearly it is not because $w^{(k)} \in \mathbb{R}^{n^{(k)} \times 1}$. This returns:

$$time = n_{mul} + n_g \iff \tag{13}$$

$$time = \sum_{k=2}^{n_{layers}} (n^{(k)} n^{(k-1)} n^{(k-2)}) + (n^{(0)} n^{(1)} 1) + \sum_{k=1}^{n_{layers}} (n^{(k)}) \tag{14}$$

When analyzing matrix algorithms, it is common to assume that the matrix is square. That is, they have the same number of rows and columns. This leads to

$$n_{mul} = n_{layers} \cdot n^3 \tag{15}$$

Assuming again that each layer has the same number of neurons, and that the number of layers equals the number of neurons in each layer, we obtain

$$n_{mul} = O(n \cdot n^3) = O(n^4) \tag{16}$$

Moreover, for the activation:

$$n_g = n_{layers} \cdot n = O(n^2) \tag{17}$$

So, the total execution time for the neural network will be in the order of

$$O(n^4 + n^2) \iff O(n^4) \because \forall n \geq 1 | n^4 + n^2 \leq 2n^4 \tag{18}$$

which, for a large value of $n$, is approximable with

$$O(n^4) \tag{19}$$

As far as the algorithm at the basis of the "sensor-based" approach is concerned, we can refer to [20], where the solution is obtained by minimizing the restricted function

$$\begin{cases} E = \sum_{i=1}^{N} \sum_{j=1}^{N} (k_{i,j} a_{i,j} (r_{i,j} - r_{i,j}^0)^2) \\ r_{i,j} = d(i,j), \end{cases} \quad \text{where } i, j \text{ are anchors} \tag{20}$$

where $N$ is the number of nodes, $a_{i,j}$ is the index variable (1 if a connection exists, 0 otherwise), $d(i,j)$ is the known distance between anchors and $r_{i,j}$ is the distance between nodes.

Minimization of this function is obtained using Sequential Quadratic Programming (SQP).

In [29], it is shown that this problem is a convex quadratic optimization problem that can be solved in polynomial type, and the complexity is about

$$O(n^3) \tag{21}$$

In essence, the time complexity of a sensor-based algorithm is an order of magnitude more efficient than a neural-network-based algorithm.

*4.2. Comparative Analysis between Power Consumption of Neural Network and Sensor-Based Approach*

Using the training data, the Matlab Code trained the network and, using the validation data, it returned the accuracy at regular intervals in the course of the training process.

It is also important to add some observations about the computational overhead introduced by the neural networks [30]. Using the MIT Deep Neural Network Energy Estimation Tool [31], we were able to quantify the energy estimation. The tool generates:

- A txt file where each row gives the estimated energy dissipation for each layer in terms of the data movement and of the three data types (input feature map, weight and output feature map) and the computation. Energy units are normalized in terms of the energy for a MAC (multiply-and-accumulate) operation (that is, 102 = energy of 100 MACs). The output total energy is the energy required to process the dataset.
- A png file which visualizes the energy estimation result. See Figure 8.

MAC operations in CONV and FC layers account for more than 99% of the total operations in a modern CNN [31] and therefore dominate both processing time and power consumption. The results in Figure 8 show that the energy consumption is comparable with that of a standard CNN.

Both of the two approaches we are comparing belong to the class of centralized systems, i.e., the systems in which the calculation takes place at the network level and not on individual nodes. In [32], it is reported that for the centralized case, the total energy dissipated per sensor is 132.7 μJ, regardless of the type of localization algorithm used. At the network level, instead, the energy complexity of the algorithm (see [33]) is obtained as

$$O^E(f_e(n)) = O_c(E_c \cdot f_c(n)) + O_p(E_p \cdot f_p(n)) + O_s(E_s \cdot f_s(n)) + O_n(E_n \cdot f_n(n)) \quad (22)$$

where $O_c(f_c(n))$ is the CPU complexity equal to the known runtime, and $O_c(E_c \cdot f_c(n))$ is the order of magnitude of runtime complexity multiplied by the energy required to operate. In the same way, $O_p(f_p(n))$ handles primary memory usage complexity, the class $O_s(f_s(n))$ characterizes the energy consumption of secondary storage and $O_n(f_n(n))$ can represent the energy complexity of network resources. In [32], the comparison of communication and processing energy using the centralized architecture is reported. It shows that communication contribution is negligible when the number of sensor nodes exceeds 50 units, so we approximate the equation this way:

$$O^E(f_e(n)) = O_c(E_c \times f_c(n)) + O_p(E_p \times f_p(n)) + O_s(E_s \times f_s(n)) \quad (23)$$

In the standard "sensor-based" case, we therefore have

$$O^E(f_e(n)) = O_c(E_c \times n^3) + O_p(E_p \times f_p(n)) + O_s(E_s \times f_s(n)) \quad (24)$$

For all Strassen-like algorithms (that is, those based algebraically on upper bounds on the rank of matrix multiplication) [34], the space usage is at most $O(n^2)$. Assuming that energy consumption on primary and secondary memory is comparable, using some experimental results of energy consumption, as in [32], and neglecting the contribution of $n^2$, we obtained this equation:

$$O^E(f_e(n)) = 0.0009271406 \cdot n^3 \quad (25)$$

In the case of the neural network, we obtain

$$O^E(f_e(n)) = O_c(E_c \cdot n^4) + O_p(E_p \cdot f_p(n)) + O_s(E_s \cdot f_s(n)) \quad (26)$$

In [35], it is shown that layer depth, epoch and batch size are variable by trail and do not affect memory consumption. Regarding the learning rate, the smaller the learning rate, the higher the memory consumption. So, we must take into account the learning rate that we used in the simulation, that is 0.0007. In this situation, the memory usage is about 2700 Megabytes. In [36], it is reported that the effective read/write energy of one bit is $\approx 10^{-9}$ J/bit = 1 nJ/bit

So, we obtain

$$O^E(f_e(n)) = 0.0009271406 \cdot n^4 + 10^{-9} \times 2.7 \times 10^9 \times n \tag{27}$$

simplifying and approximating, we obtain

$$O^E(f_e(n)) = 0.0009271406 \cdot n^4 \tag{28}$$

As we can see, the most important contribution is given by the computational process. It appears evident that the energy consumption, at the network level (Figure 9), in the case of standard architecture is an order of magnitude lower than that achieved with the neural network, although at the network-level power consumption is not an issue, while the power consumption at the sensor level, as seen before, is the same in both architectures.

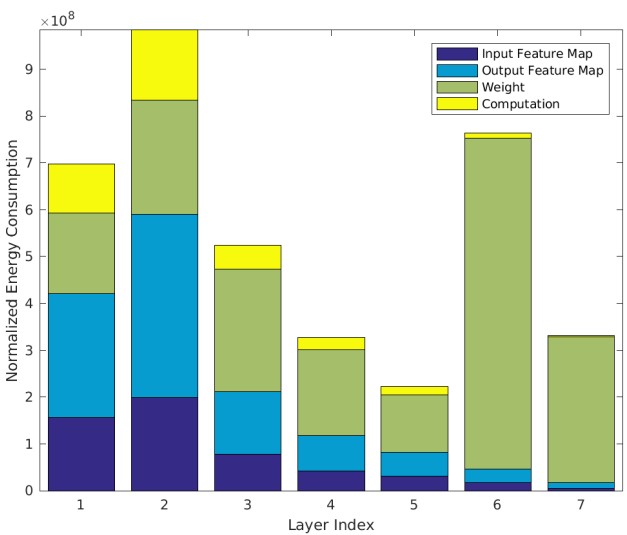

**Figure 8.** This figure shows the energy estimation result due to the introduction of the neural network.

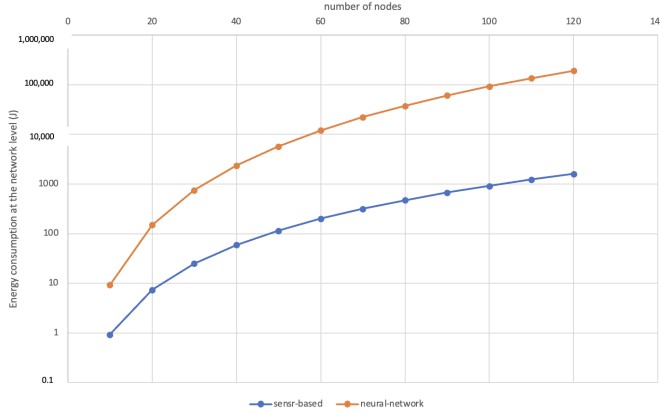

**Figure 9.** This figure shows the energy consumption at the network level for both the architectures.

## 5. Results

The results obtained show that it is possible to reach an average localization error lower than that obtained with the kNN technique. In fact, we obtained an error of less than 150 m. The keystone is linked to the ratio between the number of clusters and the mean cluster size.

$$\lambda = \frac{number\_of\_clusters}{mean\_cluster\_size} \tag{29}$$

If this ratio, named "crowding index", is greater than 5%, the accuracy is not guaranteed, because the number of clusters is too high compared with the mean cluster size. On the other hand, when the value is less than 5%, we find a very high accuracy. Accuracy is the ratio of correct predictions and the total number of classes. That is,

$$Accuracy = \frac{TruePositive + TrueNegative}{TruePositive + TrueNegative + FalseNegative + FalsePositive} \quad (30)$$

*5.1. Low Accuracy*

When $\lambda$, the crowding index, is greater than 5%, the poor accuracy in the localization measurement emerges. The results are distributed as the following example:

1.  Number of clusters: 4;
2.  Size of smallest cluster: 13;
3.  Size of largest cluster: 123;
4.  Mean cluster size: 50.500000;
5.  Median cluster size: 33;
6.  Number of points that are not part of any cluster: 9.

Another example of poor accuracy occurs in the middle eastern area of the city, where we find the following values:

1.  Number of clusters: 5;
2.  Size of smallest cluster: 10;
3.  Size of largest cluster: 67;
4.  Mean cluster size: 36;
5.  Median cluster size: 26;
6.  Number of points that are not part of any cluster: 18.

In Figures 10 and 11, we can see, respectively, the behavior of the training process and the confusion matrix in the case of $\lambda = 8$%. The confusion matrix table briefly describes the predicted outcome for the classification problem. In this case, it shows that it predicts 81% of data correctly, and 19% of the data were mislabeled in the validation dataset. In Figures 12 and 13, the behavior of the training process and the confusion matrix in the case of $\lambda = 13$% are shown, respectively. The confusion matrix returns the predicted outcome for the classification problem. In this second case, it shows that it predicts 75% of the data correctly, and 25% of the data were mislabeled in the validation dataset.

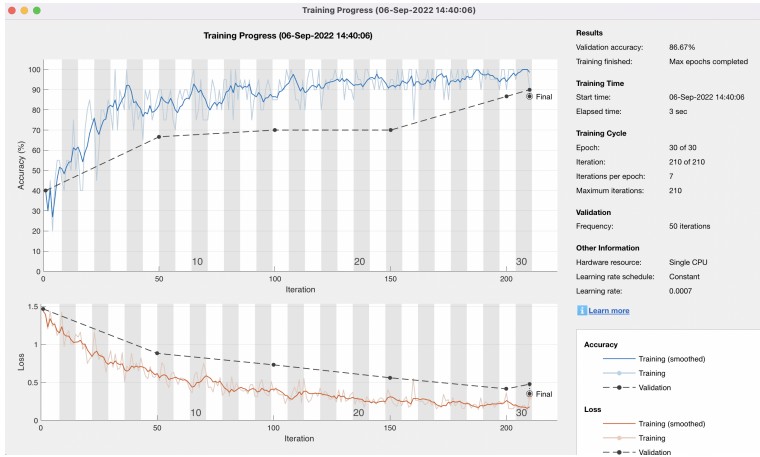

**Figure 10.** This figure shows the trend of the training progress in the case of a subarea in which the lambda value is equal to 8%.

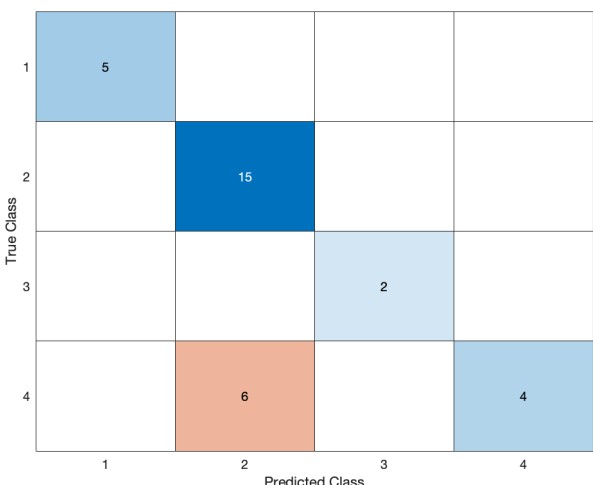

**Figure 11.** This figure shows the confusion matrix in the case of a subarea in which the lambda value is equal to 8%.

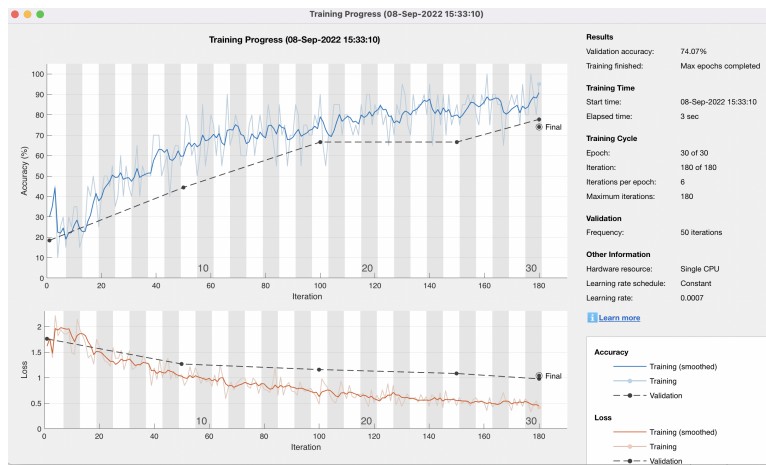

**Figure 12.** This figure shows the trend of the training progress in the case of a subarea in which the lambda value is equal to 13%.

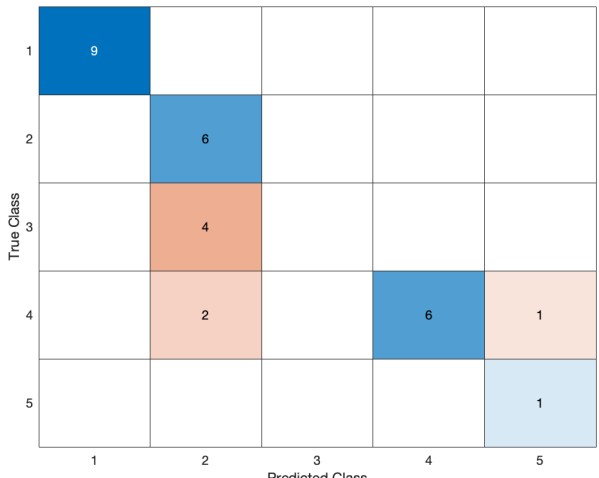

**Figure 13.** This figure shows the confusion matrix in the case of a subarea in which the lambda value is equal to 13%.

The big variance in the cluster size is linked to the type of journey made by the vehicles of the Antwerp postal service and to the different times spent in the single subareas.

*5.2. High Accuracy*

In the case of high accuracy in the localization measurement, the results are distributed as the following example:

1. Number of clusters: 15;
2. Size of smallest cluster: 16;
3. Size of largest cluster: 1175;
4. Mean cluster size: 359;
5. Median cluster size: 350;
6. Number of points that are not part of any cluster: 6.

In this case, Figures 14 and 15 show that the neural network predicts 95% of the data correctly, and 5% of the data were mislabeled in the validation dataset.

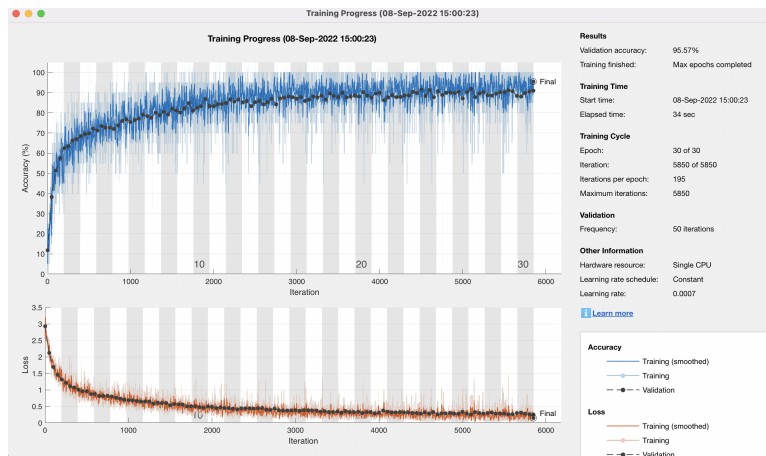

**Figure 14.** This figure shows the trend of the training progress in the case of a subarea in which the lambda value is equal to 4%.

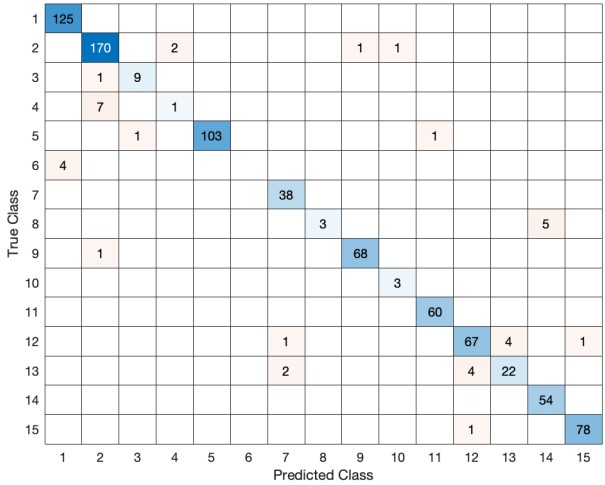

**Figure 15.** This figure shows the confusion matrix in the case of a subarea in which the lambda value is equal to 4%.

Another example of a lambda value lower than 0.05 is the following:

1. Number of clusters: 10;
2. Size of smallest cluster: 12;
3. Size of largest cluster: 1784;

4.    Mean cluster size: 299;
5.    Median cluster size: 121;
6.    Number of points that are not part of any cluster: 18.

In this last case, $\lambda = 0.03$.

In this case, Figures 16 and 17 show that the neural network predicts 95% of the data correctly, and 5% of the data were mislabeled in the validation dataset.

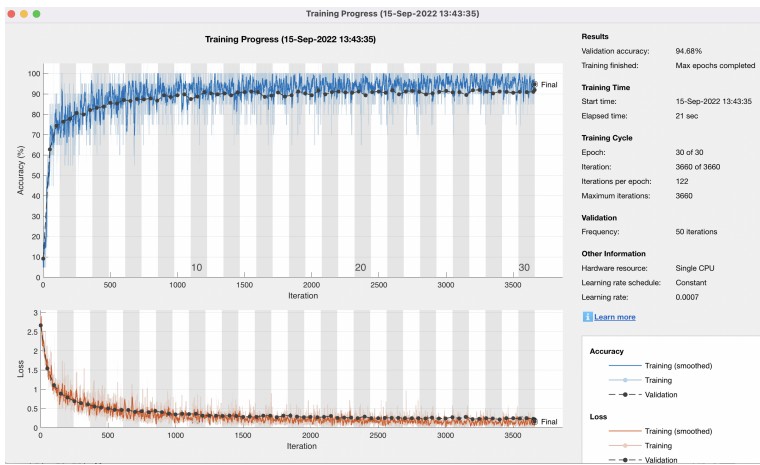

**Figure 16.** This figure shows the trend of the training progress in the case of a subarea in which the lambda value is equal to 3%.

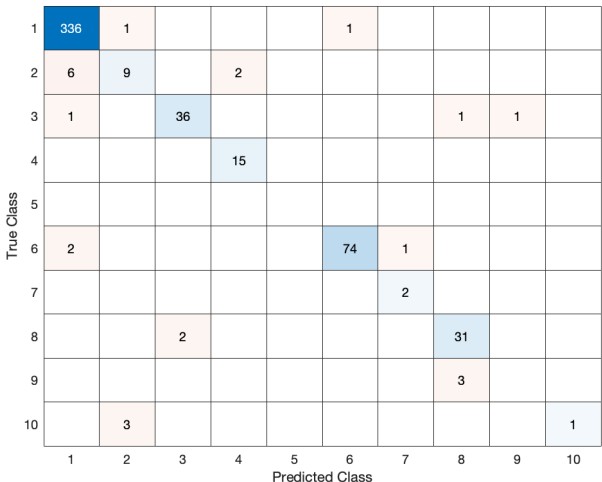

**Figure 17.** This figure shows the confusion matrix in the case of a subarea in which the lambda value is equal to 3%.

## 6. Discussion

According to the literature, the initial results of a basic fingerprinting implementation show an average location estimation error of 398.40 m for the urban LoRaWAN dataset using the standard kNN algorithm. The purpose of this work was to verify whether a lower location estimation error level could be achieved with a machine learning approach. The results of this work show that it is possible to achieve greater accuracy as long as the lambda ratio has values lower than 0.05, that is, the ratio between the number of clusters and the mean cluster size (crowding index) is less that 5%. The average position estimation error obtained with the machine learning approach is less than 150 m, and this is an important milestone for the growing relevance of the Internet of Things and location-based services. The precision can be increased up to a few meters ($\sim$5 m), but this is only possible by increasing the sensor density per space unit. Figure 18 shows the case in

which, with the neural network, we set a precision of a few meters (∼5 m). The accuracy of the measurement drops drastically to values around 55%. This phenomenon is explained by the "crowding index", which in this case assumes values of 2.16, significantly higher than the critical threshold. In Table 4, the accuracy results obtained with different values of lambda are represented. The outcome of the simulation with different lambda values confirms what was previously stated. In any case, the proposed solution is substantially at no cost, as it does not require implementations of dedicated networks but uses networks that have already been created (for example gas meters, electricity meters, water meters, etc.) simply by integrating a few lines of code on each firmware, an operation that can be performed remotely. In [37], a performance comparison of RSS- and TDoA-based location approaches in an outdoor public LoRa network is presented. The raw location estimates of the TDoA approaches outperform all RSS approaches examined, with median errors on the order of 200 m versus median errors on the order of 1250–2500 m [3]. With the present work, we instead demonstrated that, by using a neural network applied to an RSS dataset, it is possible to go far below that threshold. A completely different approach is presented in [17], where through the OwLL, a LoRa localization system, it is possible to reach an accuracy of a few meters ∼9 m. Obviously, this is a specific and proprietary solution and which by its nature has costs associated and needs a specific business case analysis.

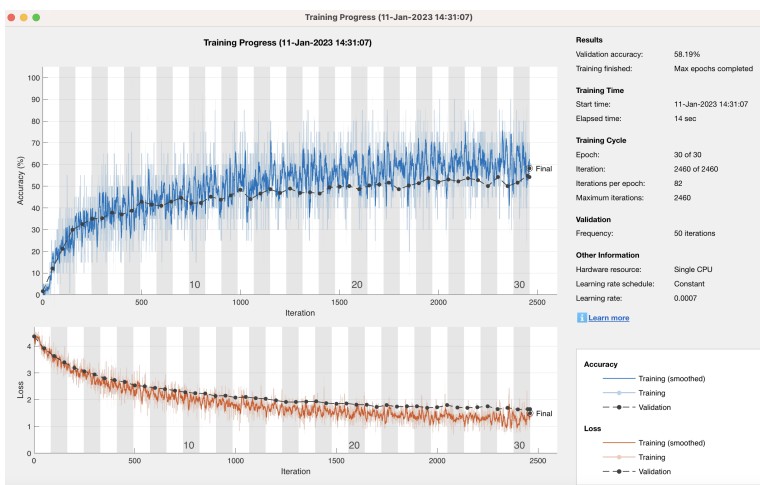

**Figure 18.** This figure shows the accuracy obtained in the case of an average position estimation error of 5 m.

**Table 4.** Comparative table of different lambda values and relative accuracy results.

| Lambda | Accuracy Result |
|--------|-----------------|
| 0.006  | 98%             |
| 0.01   | 95%             |
| 0.03   | 95%             |
| 0.16   | 84%             |
| 0.3    | 59%             |

## 7. Conclusions

A localization algorithm with an estimation error of this magnitude is suitable for many applications. Advanced systems that improve and automate processes within cities will play a leading role in smart cities. From intelligently designed buildings that collect rainwater for later use to intelligent control systems that can autonomously monitor infrastructure, the potential improvements enabled by sensor technology are immense. Ubiquitous sensing inherently poses many technical and social challenges [38]. Improved localization ensures security and up-to-dateness where needed. A study in France assumes that the average parking space search time in large cities is up to 40 min, which equates to

70 million hours or EUR 700 million in costs per year [39]. Another possible application is related to the customized signaling of escape routes in shopping centers or stadiums in case of adverse events. Once the threat has been identified, the system can guide users in a dynamic and personalized way, providing the necessary actions to keep them safe (e.g., finding the safest way out, not necessarily the closest one). Unmanned Aerial Vehicles are another interesting application set. Measurements are important both for drone motion and the performance of the drone's intended mission. The sensors used to collect the data required to operate unmanned vehicles vary widely in terms of technology, performance and the measurement accuracies to consider. The range of applications for these systems is constantly increasing in number and complexity, requiring new measurement options and methods [40]. Location-based services are a growing market. Location-based services not only bring commercial benefits, they can also improve the quality of life for people with disabilities and save lives in emergency situations. Future work will be focused on sensor localization with UAVs and, in particular, on UAV tracking techniques and cooperative control of multiple UAVs.

**Author Contributions:** Conceptualization, A.P. and M.T.; methodology, A.P.; software, A.P.; validation, M.T.; formal analysis, A.P.; investigation, M.T.; writing—original draft preparation, A.P.; writing—review and editing, M.T. All authors have read and agreed to the published version of the manuscript.

**Funding:** This research received no external funding.

**Institutional Review Board Statement:** Not applicable.

**Data Availability Statement:** The Matlab Code used for this research is available online: https://github.com/apirodd/LoRaWAN (accessed on 23 November 2022); Antwerp LoRaWAN Dataset is available online: https://zenodo.org/record/3342253/files/lorawan_antwerp_2019_dataset.csv?download=1 (accessed on 23 November 2022).

**Conflicts of Interest:** The authors declare no conflict of interest.

**Sample Availability:** The Matlab Code is available at the following repository: https://github.com/apirodd/LoRaWAN (accessed on 23 November 2022).

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
