# Peer review of "Machine Learning Applied to LoRaWAN Network for Improving Fingerprint Localization Accuracy in Dense Urban Areas"

_2673-8732, doi:10.3390/network3010010_

Round 1
Reviewer 1 Report
1. The author needs to ensure the quality of the Figures in the paper. For example, Figure 1 is not clear and has shadows, Figure 6 is not complete (fire-squeeze..., fire2-rele_exp...), and Figure 4,5 are not self-explanatory.
2. The captions of the Figures in Section 5 are confusing and I hope the author can revise the explanation.
3. In Section 5, the author only shows the results of two different choices of lambda. In Section 6, the author proposes “The results of this work show that it is possible to achieve greater accuracy as long as the lambda ratio has values lower than 0.15, that is the ratio between the number of clusters and the mean cluster size (crowding index) is less that 15%.” I think the evaluation results shown in Section 5 cannot draw the conclusion in Section 6 and the author need to explanation the relationship between the result and the discussion.
4. It is hoped the author can evaluation the performance of the system under more different choices of lambda (0%, 5%, 10%, 15%, … 100%) to show the best configuration of the Neural Network.
Author Response
- Following your suggestions: figure 1 has been enlarged to make it clearer; figure 6 has been replaced with one showing the whole architecture of the neural network. The captions of figures 4 and 5 have been modified making them more self-explanatory.
- Following your suggestions: the captions of the figures in section 5 have been completely revisited, trying to better explain their contents.
- Following your suggestions: we've added other 2 examples of different value of lambda, showing with pictures the accuracy results.
- Following your suggestions: We've also added a comparative table of different lambda values ​​and relative accuracy results
Reviewer 2 Report
This paper focuses on enhancing localization accuracy using LoRaWAN and machine learning techniques. The proposed solution is energy-efficient compared to existing methods, e.g., GPS. I have several comments which should be addressed before accepting this manuscript for publication.
1. In the manuscript, the sentences are overly long and look like paragraphs. Therefore, the technical writing needs revision to make the manuscript readable for the potential audience. For example,
“LoRaWAN technology provides a two-way communication, but the transmission from node (also known as motes) to gateway (also known as concentrator or base station) or Uplink message is the most frequent one compared to that from gateway to node or Downlink, since usually the purpose of the nodes is to collect data and then send them to the Network Server and then to the Application Server.”
Similarly, please revise the other long sentences.
2. Please replace "LoRaWAN motes" to LoRaWAN end-devices, which is a widely used and known term in LPWA community.
3. Please note that LoRa is a modulation and LoRaWAN is a protocol. The communication between node and gateway follows LoRa modulation. Please correct “the nodes send Uplink messages to the gateways in radio frequency through LoRaWAN modulation.” -> LoRa modulation.
4. “The Network Server also manages the transmission speed”. Please change “transmission speed” -> data rate or bit rate
5. Manuscript needs additional references to support the technical details. For example, “the TTN Network Server uses the 20 most recent 67 Uplink messages [REFERENCE?]”
6. “spread factor” -> spreading factor (SF). In figure 1, why 3G? Why not 4G or 5G?
7. “The spread factor controls the frequency of the chirp”. Please note that SF does not control the frequency of the chirp. SF represents the number of bits a chirp. There’s a trade-off between the SFs and bit rate, network coverage, energy consumption. Overall, section 2 “LoRaWAN standard” should be revised to include accurate information.
8. Any justification for the big variance in cluster sizes?
9. The information about the communications parameters is missing. For example, in the dataset, which spreading factor was used to send data from cars to the gateway? What was the signal bandwidth? What was the communication carrier frequency? What was the transmission power?
10. The current conclusion section is weak and short.
Author Response
- Following your suggestions the sentences have been shortened to make the manuscript more readable.
- Following your suggestions, the word motes has been replaced by devices.
- The oversight about the LoRa modulation has been fixed.
- Following your suggestions "transmission speed" has been replaced with bit-rate.
- The reference has been added.
- Following your suggestions, figure 1 has been modified adding 4G and 5G.
- We don't address the trade-off between the SF and bit rate, network coverage, energy consumption in this paper.
- Following your suggestions it has been added in the manuscript the reason of the big variance in the cluster size.
- Apart from the spreading factor, the other transmission parameters are not reported in the dataset (University of Antwerp).
- Following your suggestions, conclusion has been reinforced.
Reviewer 3 Report
Title: Maching Learning Applied To LoRaWAN Network for Improving Fingerprint Localization Accuracy in Dense Urban Areas
Strengths:
- The problem presented in the paper is interesting.
- The paper is, for the most part, well-written and easy to read.
Weaknesses:
- The paper is missing several technical details.
- The authors did not discuss the computational overhead of the neural network.
- The authors made a strong claim regarding the need for accuracy in LPWAN.
- Several existing works are not cited, which achieve an accuracy of 10s of meters.
Comments:
Accurate localization in LPWAN is still an open research problem. This paper presents a good effort to address such a problem. However, the paper has several drawbacks.
The authors did not provide sufficient background on RSS-based localization and its limitation. RSS-based localization suffers strong signal attenuation, limiting the accuracy of the proposed approach.
Utilizing neural networks increases the computational overhead. The paper did not discuss the impact of such overhead. Additionally, what would be the expected delay of localization?
Since LoRaWAN localization is the interest of this work, the authors must include the existing efforts on LoRWAN localization which achieve meter-level accuracy [1, 2].
Overall, I believe the paper is addressing an interesting problem. However, it lacks an extensive analysis of the proposed technique.
References:
[1] Bansal, Atul, Akshay Gadre, Vaibhav Singh, Anthony Rowe, Bob Iannucci, and Swarun Kumar. "Owll: Accurate lora localization using the tv whitespaces." In Proceedings of the 20th International Conference on Information Processing in Sensor Networks (co-located with CPS-IoT Week 2021), pp. 148-162. 2021.
[2] Plets, D., et al. Experimental performance evaluation of outdoor tdoa and rss positioning in a public lora network. In IEEE IPIN (2018).
Author Response
- Following your suggestions some more insights about the RSS-based localization and its limitation have been added.
- Following your suggestions some considerations and results about energy overhead due to the introduction of the neural network have been added.
- Following your suggestions some more insights about the existing efforts on LoRWAN localization which achieve meter-level accuracy [1, 2] have been added.
Reviewer 4 Report
The paper is very well written and very clear in the presentation of results and conclusion.
I believe that there are a number of studies that have re-used this dataset and one weakness is that there didn't seem to be a significant comparison between this approach and others that have previously been reused.
The portion of the paper where you discuss the application of this localisation based technology could be significantly strengthened. Self driving cars are unlikely to use it, but there are other low power environmental sensors that you could have discussed.
The limitation where significant training, on a specific dataset, is required is a significant limitation and requires some greater discussion.
Author Response
- With regard to the observation relating to the comparison with other studies based on the same dataset, the objective of the paper is precisely to introduce a different analysis methodology that can provide appreciable results.
- Following your suggestions, in the conclusions paragraph, some more observations about the applicability of fingerprint localization with high accuracy have been added.
- Following your suggestions, it has been detailed that the analysis must be deepened using some other dataset in order to generalize the results.
Round 2
Reviewer 1 Report
Figure 6 does not need to occupy so much space and the information contained can be more dense.
No other comments.
Author Response
Following your suggestion the dimension of Figure 6 has been reduced.
Reviewer 2 Report
My concerns are addressed, and the reviewer has no further comments. The paper can be accepted.
Author Response
Thank you for the helpful comments.